# How does the dental benefits act encourage Australian families to seek and utilise oral health services?

Peivand Bastani[1]*, Reyhane Izadi[2], Nithin Manchery[1], Diep Ha[1], Hanny Calache[3], Ajesh George[4,5,6], Loc Do[1]

1 School of Dentistry, The University of Queensland, Brisbane, Queensland, Australia, 2 School of Health Management and Information Sciences, Shiraz University of Medical Sciences, Shiraz, Iran, 3 La Trobe Rural Health School, Australian Centre for Integration of Oral Health (ACIOH), La Trobe University, Melbourne, Australia, 4 Australian Centre for Integration of Oral Health (ACIOH), School of Nursing and Midwifery, Western Sydney University, Penrith, Australia, 5 Ingham Institute Applied Medical Research, Liverpool, Australia, 6 School of Dentistry, The University of Sydney, Camperdown, Australia

* p.bastani@uq.edu.au

**Data Availability Statement:** All relevant data are within the paper. All the data are documented and presented in Tables 1–3.

**Funding:** The author(s) received no specific funding for this work.

## Abstract

### Background

This study aimed to analyse the content of the Dental Benefits Act 2008 as a foundation for the Child Dental Benefits Schedule (CDBS) to determine how the Act encourages Australian families to seek and utilise oral health services.

### Methods

This was a qualitative narrative document analysis conducted in 2022. Data was collected by searching formal websites for retrieving documents that reported the Australian Dental Benefits Act. The eligibility of the retrieved documents was assessed based on authenticity, credibility, representativeness, and meaningfulness of the data. A seven-steps procedure was applied for framework analysis.

### Results

The content of the Dental Benefits Act 2008 provides directions on the three categories of operational, collective, and constitutional rules. Operational rules at the level of oral health providers and the population, as the service end users, can be demonstrated as rules in use in a mutual interaction with the collective and constitutional rules. The consequence of governing the rules at the community level can easily define how the oral health services are provided and utilised. The response is sent to the government level for better regulation of oral health service delivery and utilisation. Then, with interaction and advocacy with the diverse range of stakeholders and interdisciplinary partnerships, with community groups, non-government sectors and councils, the rules can be transformed, adopted, monitored, and enforced. Another mechanism of response has occurred at the providers' and users' level and to the operational rules to community groups and stakeholders via advertising and promoting the utilisation and provision of oral health services.

**Competing interests:** The authors have declared that no competing interests exist.

## Conclusion

This study integrates the perspective of politicians with those of policy makers to reconsider the role and significance of the rules based on the triple collaborations among oral health users and oral service providers, the community, and the stakeholders as well as the government. A comprehensive attention is still needed in future revisions of the Dental Benefits Act 2008 according to the contextual factors, socioeconomic and geographical attributes of the population for better implementation of de facto rules and more effective outcomes of the interventions. It is recommended that further research be undertaken utilising a mix-method approach for a holistic view prior to further revisions of the Act or proposal of probable upcoming schemes.

## Background

Oral health is integral to overall health and well-being of the population. This includes physical, psychological, emotional, and social domains [1]. Due to the multidimensional impacts of oral diseases, which include chewing problems, nutritional deficiencies and weight loss, irritability, insomnia, low self-esteem, and decline in social confidence and performance [2], oral health is considered, by World Health Organization (WHO), to be an essential part of general health interventions and practices [3]. To achieve this goal, global policymakers have tried to develop strategies to increase the population's access to dental services along with improving the population's oral health literacy, behaviours and service utilisation based on their contextual conditions, infrastructures, and facilities [4]. For instance, the National Institute for Health and Care Excellence (NICE) released policies and interventions for the improvement of the population's oral health. These strategies are included in local health and wellbeing policies, public service environment and workforce policies, nursery services and school policies, and policies identifying high-risk groups for poor oral health [5].

In Australia, the provision of oral health services is either via a fee for services mechanism that operates within the private sector or via the provision of oral health services through State and Territory Government public sector for eligible population groups, which includes children and those with low socioeconomic status. Although there is a policy shift toward preventive services and early intervention programs among children and teenagers, two federal dental programs based on national laws and legislations were implemented over the period 2007 to 2013. These were the Chronic Disease Dental Scheme (CDDS) from 2007 to 2012 and the Medicare Teen Dental Plan (MTDP) during 2008 to 2013. The evidence related to the implementation of these programs indicated that these programs were not effective and had low utilisation rates particularly in rural and remote areas [6]. For instance, the evidence around the utilisation of MTDP in New South Wales (NSW) during 2008–2010 revealed concerns related to the program, which include lack of uptake, equity of vouchers' uptake, availability, and willingness of the providers, particularly among rural and regional areas, to accept vouchers and insufficient support for providing follow-up care in the private sector [7]. These concerns may have contributed to the reasons for closure of the MTDP program by the Australian Federal government at the end of 2013. Following the trend and history of the national public programs, the Child Dental Benefits Schedule (CDBS) was implemented in 2014 to provide clinical dental treatments to eligible 2–17 years-old children and teenagers up to a payment limit of $1000.00 over two consecutive calendar years [8]. Since 2022, the eligibility has been increased to children 0–17 years with a payment limit of $1026.00 over 2 consecutive

calendar years. Although a wider range of dental treatment services was included in the CDBS compared with the MTDP, low utilisation rate (23%) still exists [9]. Recent evidence by Stormon et al. (2022) indicates an increase of only 8% in access to dental services among low-income households through the CDBS [8].

Given the challenges for increasing the utilisation of the CDBS it is important to have a big picture of the acts, laws and legislations tabled and released by the parliament as the root and background of such schemes. Reviewing this information is important before policy makers make a conclusion based on the evaluation of such schemes and explore other avenues to redesign new interventions or continue with existing schemes, The Dental Benefits Act 2008 (the Act) which commenced in June 2008 is one of the critical legislative frameworks for the provision of dental benefits in Australia. The Act has had four reviews over the previous decade. The first review which was tabled in parliament in 2010, mainly focused on the attainment of the purpose and the administration of the Act related to the MTDP (10). The second review in 2012, concentrated on the operations and administration of the Act related to the MTDP. The third review led to the closure of the MTDP and the establishment of the CDBS and finally, the fourth review in 2019, considered the operation and administration of the Act, in relation to the CDBS [10].

Being successful with the purpose of implementing such an Act, via the stated schemes is not only related to the clarity, power, coverage, and legal and administrative aspects of establishing the rules by the government which are labelled as constitutional rules but also depends on both collective rules and operational rules. In other words, constitutional rules, as defined by the government, can affect the formal and informal aspects of collective rules. Constitutional rules are those based on approved formative constitutions related to fundamental principles of the government's authority which define the interpretation and application of the powers, rights, and freedoms. Collective rules are set by the community groups and can affect the operational rules via their impacts on the demand and supply side of the oral health market including community groups, religious groups, women or other vulnerable parties on the demand side and the professional groups (oral healthcare providers), government and local councils on the supply side. The operational rule is where individuals (users and providers) make choices and where rules-in- form (de jure rules) transform into rules-in-use (de facto rules) [11]. For more clarity, de jure rules (rules-in-form) describe those practices that are legally recognised, regardless of whether the practice exists in reality while de facto rules (rules-in-use) describe those practices that exist in reality, even though they are not officially recognised by laws. Many determinant factors can be considered from the users' perspective while the market forces defined by demand and supply rules can also be influential in day-to-day decisions of the population to utilise the oral health services. These are all considered as operational rules that can be affected directly by collective rules or both directly and indirectly by constitutional rules. Context including this rule in use, socioeconomic and geographic attributes, can affect the whole collaborations among these triple rules. Fig 1 illustrates this conceptual framework.

In order to make an understandable and recognisable language between politicians, policy makers and healthcare providers in the area of oral health, we plan to review the content of the Act and its related reviews, considering the government not only as a singular entity and steward of the community's health but also as the enforcer or the body that implements the rules directed by the acts and at the same time organised by authorities and informal groups and stakeholders [11]. Such a framework can present a rational combination among the directions of the acts set by politicians and the governors' rules in oral health. So, this study aimed to analyse the content of the Dental Benefits Act 2008 to determine how this Act encourages Australian families to seek and utilise oral health services.

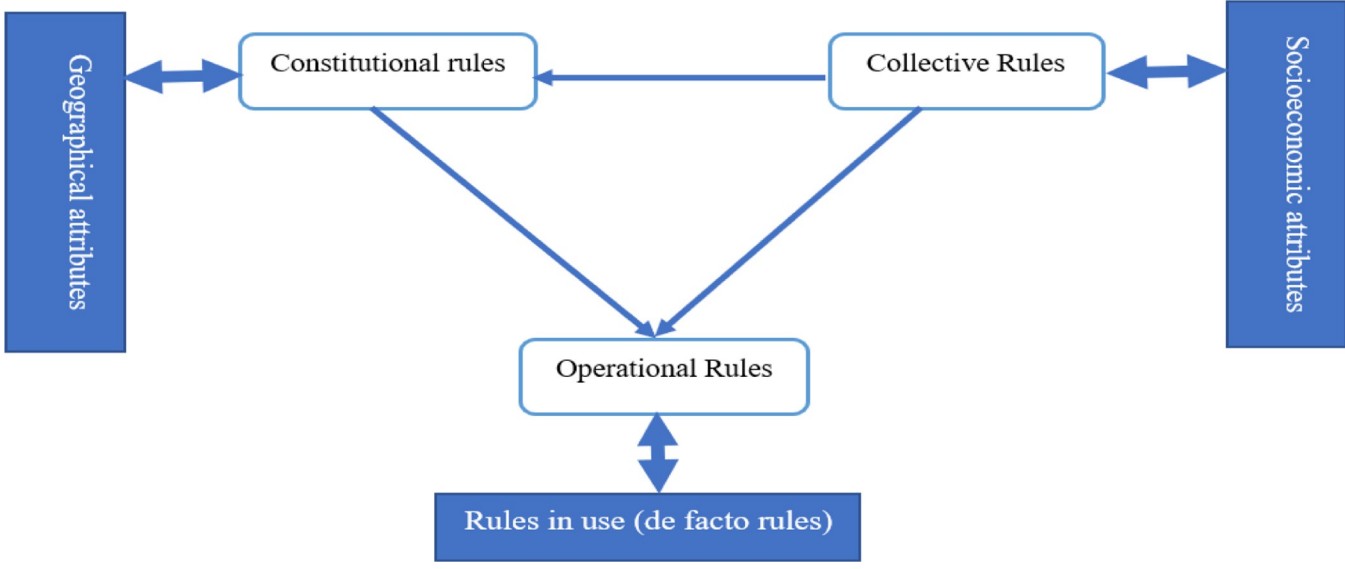

**Fig 1. Conceptual framework of the analysis.**

## Method

This qualitative narrative document analysis, was conducted in 2022, using a framework analysis approach. The full content of the Australian Dental Benefits Act 2008 and the related reviews were considered as the qualitative data. As mentioned by two of the qualitative experts Gupta (2015) and Krippendorf (2012), official governmental documents, guidelines and directives, programs, policies and periodic reports, and the content of laws and legislations can be analysed for their contents as a rich qualitative source of data [12, 13].

### Data collection

Data was collected by searching the formal websites for documents related to the Australian Dental Benefits Act 2008. The whole content of the Act as well as the four versions of its revision were retrieved as the public free documents. The eligibility of the retrieved documents then was investigated and approved according to the suggested criteria by Scott (2014) including authenticity, credibility, representativeness, and meaningfulness of the data [14].

### Data analysis

After retrieving the data and inclusion of the documents according to Scott criteria (2014), we have used the procedure suggested by Gale et al. (2013) for framework analysis including the following seven-steps [15]:

- *Transcription*: This first step was not actually done due to the existence of a portal document format (pdf) of the Acts files. So, the content of the files was saved in a word format with large margins and sufficient line spacing for later coding and note taking which occurred in the following steps.

- *Familiarisation*: To be familiar with the content of the Acts, all the files were read several times line by line. All the impressions or important issues were noted in the margins of the pages.

- *Coding*: at this point, after familiarisation with the whole texts, the parts related to the research question were highlighted and a paraphrase or label was selected which can describe well the interpretation of the text. These labels demonstrated the initial codes. This was an open coding inductive approach with the aim of summarising the whole data towards the main concept.

- *Developing a working analytical framework*: In this step the initial codes extracted from the content of the Act by one of the researchers (RI) was reviewed by the team and the code labels were discussed, compared, and agreed to by the team. Then the initial codes were categorised to make the final codes apply to a tree diagram to develop an initial analytical framework.

- *Applying the analytical framework*: MAX QDA version 10 was used in this step as a Computer Assisted Qualitative Data Analysis Software to determine the relationships among the initial codes, final codes, and upper-level categories. These categories first create the sub-themes and synthesised the sub-themes to achieve the main themes.

- *Charting data into the framework matrix*: To integrate the inductive open coding approach with a deductive conceptual framework which best fits the emerging concepts, the triangle of rules proposed by Abimbola (2020) was applied [11] at this stage. Then the data was charted into a table based on the developed framework (Fig 1).

- *Interpreting the data*: In this last step, the developed framework including the main themes and related sub-themes were described and interpreted in accordance with prior concepts and new ones emerging from the data.

## Data robustness and trustworthiness

Four criteria suggested by Lincoln and Guba (2017) were applied to ensure data analysis robustness and trustworthiness including credibility, transferability, dependability, and confirmability [16]. To achieve credibility, prolonged engagement with the data occurred through a familiarisation process. The coding process was initially done by RI and NM and finalised by PB through a consensus meeting. An external check was also done by a qualitative research expert outside of the research team to review the whole research process, which was followed by a debriefing session. Presenting a thick description of the findings was then implemented to achieve transferability. To ensure dependability, the whole research process was documented clearly in detail and with a logical and traceable manner. And finally, to get confirmability, the interpretation of the findings was described objectively and neutrally to show that all the findings were derived from the initial data. Furthermore, the reflexivity of the qualitative analysis was assured by the research team collaborating with the qualitative research experts (PB, AJ, HC).

## Ethical considerations

Data was analysed by two of the authors (RI and PB) with no conflict of interest against the topic. Ethics clearance was not required.

## Results

The content of Dental Benefits Act 2008 provides directions on all three categories of operational, collective, and constitutional rules. Tables 1–3 demonstrate the main themes and sub-themes identified from analysing the content of the Act in each category of the rules.

**Table 1. The main and sub-themes related to operational rules.**

| Theme | Sub theme | Final code |
|---|---|---|
| Consequences of the rules governing how the service is used and provided | **General rates of use and delivery of services** | Low CDBS[a] usage rate as a reflection of poor promotion |
| | | Increase the number of active providers, service recipients, benefits, and CDBS performance indicators |
| | | Low rate MTDP[b] plan uses in eligible teenagers and lack of continuous increase rate in using vouchers |
| | | MTDP: Issuance of coupons and services per coupon per year for eligible individuals |
| | | MTDP overall success, based on goals and performance indicators |
| | | MTDP: weakness in Actual benefits paid, utilisation target, actual utilisation allocation |
| | **Utilisation and service delivery rate based on specific strata/ geographical areas and public and private sectors** | CDBS: The similarity of the usage rate in major cities and regional areas, sharp decrease in the usage rate in remote areas |
| | | CDBS: Lowest service usage in SA and Tasmania and highest in Northern Territory and WA |
| | | CDBS: A reasonable proportion between the eligible population and the rate of service use in each SEIFA[c]decile |
| | | CDBS: Lower use rates for aboriginal children than non- aboriginal children, despite higher population eligibility |
| | | Injury to needy patients, especially distant children with irregular visitation patterns due to the exclusion of CDBS for in-hospital dental services |
| | | CDBS: Revealing private sector dominance in service delivery in Australia-wide |
| | | CDBS: Uncertainty in access to services due to administrative and financial problems in public sector |
| | | CDBS: The need to expand the private sector in NSW and the public sector in SA and Tasmania, and cooperation between states to understand the different use rates and remove restrictions |
| | | MTDP: Similarity in bulk billing rates in major cities, inner and outer areas, but higher bulk billing rates in remote areas, and higher uptake rates in more affluent areas due to more providers |
| Respond to collective rules: norms for service use and delivery | **Advertising norms affecting the use and delivery of services** | CDBS: The need to increase advertising to clarify the importance of oral health in children |
| | | CDBS: Use of other advertising methods (campaign, brochures, radio, video in public centers, dental offices, ethnic press, school newsletters, social media) along with eligibility notifications |
| | | Need to revise the eligibility notifications to ensure better understanding of recipients and more attractiveness, poor promotion, and the need for more serious marketing in CDBS |
| | | MTDP: Managing family expectations using appropriate communication materials and easy-to-understand content to increase the rate of absorption and use |
| | | MTDP: Public advertising of plan through brochures, websites, social media at the beginning and throughout the year |
| | **Promoting as a marketing mix for specific groups** | the need to focus promotion among needy and special groups (communication channels, Follow-up letter) |
| | | MTDP: Advertisements through letters to dentists and families with qualified teenagers, advertisements focused on specific groups (school newsletters, ethnic press, community radio) |
| | | Focus on promoting MTDP amongst at-risk groups by community leaders, charity groups, and dentists |
| | | MTDP: Clarification and explanation of the value of vouchers for high-risk or impassable groups |
| | | Eligibility notifications as the most important method of promotion, |
| | | The need to expedite the issuance of vouchers and improve its branding as the strongest communication element |

*(Continued)*

**Table 1.** (Continued)

| Theme | Sub theme | Final code |
|---|---|---|
| Respond to regulation on service use and delivery | **Eligibility Requirements** | Condition for using Medicare Teen Dental Plan, eligibility in terms of age (12–17) and mean test |
| | | MTDP: Interpretation of the means test to determine of eligible teenagers12-17years old in "FTB-A"[d] recipient families or "Youth Allowance or Abstudy" recipient families |
| | | Condition for using CDBS, eligibility in terms of age or member of the eliminated needy groups but recommendations to reduce the age of eligibility due to the importance of dentistry in one to two years |
| | **Service utilisation and provision subject to eligibility of individuals** | MTDP: Clarification of the issue of not providing voucher physically during the visit, no need for physical voucher, verification of eligibility by contacting Medicare by dentists |
| | | CDBS: check eligibility notifications by phone and net/no need for physical form, adequate justification of dentists and emphasis on the need to check some items before providing the service (eligibility, number of benefits left) via SMS |
| | **Legal scope of service delivery and legal restrictions** | Step-down fees for certain services more than one service per day in CDBS (fissure sealants, five extraction items) |
| | | CDBS: the evidence of need, clinical efficacy, and safety (introducing new items) |
| | | Restrictions aimed at providing more appropriate services and benefits to patients |
| | | Exclusion of CDBS dental benefits for in-hospital dental services, the need to remove limitation to prevent harm to needy patients using the Commonwealth and state budget allocated for this issue. |
| | **Rate of service utilisation and provision based on geographical area / specific strata** | Calculation of Accessibility/Remoteness index (including eligibility and utilisation) based on geographical standards, the residence of many eligible people in major cities and remote areas |
| | | Calculation of SEIFA index (based on geographical areas), with the aim of determining the eligible children with the highest financial needs in each state and comparison of states |
| | | Identify areas based on target population and utilisation rate, target increasing utilisation rates in the Northern Territory |
| | | Identify the overall rate of use of CDBS and the number of services provided by the public and private sectors based on geographical areas, |
| | | Identification of aboriginal and Torres Strait Islander children through Medicare Indigenous Voluntary Identifier, comparison of eligibility and use between indigenous and non-indigenous |
| | | sharp decrease in the usage rate in remote areas due to lack of dentist, lack of fluoridated water supply and other factors |
| | **Service provision and utilisation rate based on the type of bill** | Two types of claim models for CDBS: bulk billing with potential implications for accessing services & patient billing as a deterrent for certain patient groups |
| | | Payment of plan benefits in two ways, a large part in the form of bulk bill, and the rest in the form of private bill |
| | **Service provision and utilisation subject to financial and treatment consent** | Complete and register the signed form of financial & treatment consent of eligible patients: 'Bulk Billing Patient Consent Form' on the first day, and 'Non-Bulk Billing Patient Consent Form' per visit |
| | | Exemption of the provider from obtaining informed financial consent in certain circumstances according to the Ministerial Guidelines |

[a] CDBS: Children Dental Benefit Schemes

[b] MTDP: Medicare Teen Dental Plan

[c] SEIFA: Socio-economic indices for areas

[d] FTB-A: Family Tax Benefit-Part A

**Table 2. The main and sub-themes related to collective rules.**

| Theme | Sub theme | Final code |
|---|---|---|
| From the level of the constitution to the enforcement of collective rules | **Transform and adopt rules from constitutional level** | Coordination of the DBA[a] 2008 and the HIA[b] 1973 on penalty & recovery arrangements, and amendments, but difference between the two in the index of access to benefits |
| | | Amendment (2018) aimed at aligning DBA 2008 and HIA 1973 on alignment of administrative arrangements, strengthening debt recovery (bulk-billing), the garnishee arrangements (not bulk-billing) |
| | | DBA, a framework for determining payments, entitlement, protection & disclosure information, issuance of vouchers, provisions on assignment of benefit & providing false information |
| | | Rules 2008 /2009 relating to the MTDP[c] operating framework and Rules 2013/2014 relating to the CDBS[d] operating framework |
| | | CDBS, as determinant of the eligible providers/children and services, how to advertise and promote the program, and conditions for payment (modeling CDBS payable benefits based on ADA[e]) |
| | | CDBS's main purpose: provide payable benefits for 'basic dental' services, identify CDBS items in the ADA dictionary using the additional two-digit prefix |
| | | The need to create an alternative process to reduce the administrative burden on access to benefits in case of ambiguity the patient's eligibility when receiving service (Clarification of extenuating circumstances in the Ministerial Guideline) |
| | | Defects in the mean test method, eliminating some needy teens based on this test, and need for group consultation with relevant organisations to expand the eligible conditions of teenagers with the aim of covering the needy groups eliminated by the means test |
| | | Administrative and eligibility requirements in the Rules (2008–2009); Designation of dental providers, eligible people for a voucher, duration and conditions required to issue the vouchers, information required to complete the account form, conditions without the need to issue a coupon |
| | | Eligible people & dental providers / accounts/invoices / Rules regarding the payment time of each service / Patient consent form rule in consent to treatment/ patient consent form rule in financial consent/ Requirement to keep clinical records (4years) |
| | | better panel understanding of the social and geographical conditions of vouchers recipients and users using SEIFA[f] index analysis, data on vouchers and services, and data related to unallocated groups, and data on receipt and use of vouchers based on bill type and geographical area |
| | | Draw charts and bills based on the "Australian Schedule of Dental Services and Glossary" as a generally coding system, and assign a three-digit code to clinical items or procedures in the ADA schedule |
| | **Monitoring and enforcement of collective rules** | Funds relating to the payment of DBA through a special appropriation |
| | | Dental Benefits payable for annual preventive dental check-ups through a dedicated unit number |
| | | Dedicated use of the 'voucher' terminology in MTDP rules, but removing it in advertising and replacing it with notification of eligibility |
| | | CDBS: Voucher validity from the time of issue until the end of the calendar year, and applying retrospective eligibility to covering expenses before issuing coupons in the same year, but existence of administrative challenges in this method |
| | | Financing under special appropriation in CDBS, and the upward trend of its expenditure from the beginning of the plan |
| | | CDBS: Payment of plan benefits based on the claim processing date |
| | | MTDP: Providing preventive dental checkups in the form of bulk bill in public dental clinics |
| | | Incremental rate of projected administered budget under Medicare Teen Dental Plan (2008–2015) |
| | | Providing the eligible verification information by Centrelink/DVA[g], matching it with Medicare information for vouchers issuance and sending bulk vouchers at the beginning of the year and sending new vouchers monthly from March via mail-out |
| | | CDBS: providing eligibility information by Centrelink/DVA, matching with Medicare information for issuance of notification of eligibility, sending eligibility notifications letter to families, teenagers, and Approved Care Organisations at the beginning of the year and every two weeks (mail-out or myGov) |
| | | Behavior changes in teenagers and improvement of oral health consequences due to proper management of MTDP and problems like lack of motivation of dentists due to fixed cost structure |
| | | Systemic problems for dentists, inability to analyse data, patient uncertainty about services received during the examination, other problems (shifting between public and private sector, the plan's lack of responsibility for ongoing treatment, the need to ensure non-examination by students) |
| | | CDBS: The requirement to obtain a second "signature" from the guardian for electronic billing/solve the issues by amending bulk billing payment service rules and redesigning the consent |

[a] DBA: Dental Benefits Act

[b] HIA: Health Insurance Act

[c] CDBS: Children Dental Benefit Schemes

[d] MTDP: Medicare Teen Dental Plan

[e] ADA: Australian Dental Association

[f] SEIFA: Socio-economic indices for areas

[g] DVA: Department of Veterans'Affairs

**Table 3. The main and sub-themes related to constitutional rules.**

| Theme | Sub theme | Final code |
|---|---|---|
| Legitimise the role of collective level of governance | **Power reference and steward authority** | The power of the Minister of Health to determine the Dental Benefits Rules using the legislative instruments to create an operational framework for the payment of dental benefits under this Act |
| | | CDBS[a] management by the Department of Human Services using information about eligibility, providing eligibility information by Centrelink/DVA[b], matching them with Medicare information, issuance of notification of eligibility based on the degree of compliance |
| | | Authorisation of states to provide access to CDBS services by Minister for Health (granting access or re-access), with the aim of providing services in areas without private sector activity and permanent access to CDBS with a long-term plan |
| | | MTDP[c] plan management by the Department of Human Services using information about eligibility |
| | | MTDP: Introduction of "RPDs"[d] by the states, preventive checkups by them, payment of RPDs -related benefits to a state bank account with income tax deduction and possibility of providing dental services by one state on behalf of another state |
| | **Rules and plans: creation, modification, enforcement, and external monitoring** | Designing the DBA[e] 2008 based on the HIA[f] 1973, and introducing the Health Legislation Amendment (2018) to further align HIA 1973 and DBA 2008, |
| | | Dental Benefits Rules 2008–2009 with the aim of determining the requirements of the MTDP and to establish a new DBS[g] |
| | | The ACT 2008 competence in designing a framework for the payment of dental benefits, its competence in the proper administration of the Medicare Teen Dental Plan, and finally the success of this plan (Examining these issues as the task of the committee (panel) |
| | | Limited implementation of the DBA, to date (2019) |
| | | MTDP role through the provision of preventive services and examinations in the form of vouchers (up to a benefit of $ 150 & the increasing rate of benefits over time) in public and private centers |
| | | Replacing the 2009 Rules with the 2013 Rules, thus ending the MTDP and starting the CDBS, with the increase in the acceptable age range, the number of eligible people, the range of services, higher profit margins, and the use of specific item numbers for each service |
| | | Creating CDBS, for eligible children aged 0–17 and benefit level has been increased $1026.00 over a two-year period for essential preventive services to improve oral health by creating habits in the long run, targeting Commonwealth expenditure on Children's oral health, building a national system for pediatric dentistry |
| | | Modeling of payable benefits in CDBS plan, based on ADA[h] |
| | | Patterned from Chronic Disease Dental Scheme, but the need to adapt CDBS based on Veterans' Affairs 'step-down fees' model to improve clinical effectiveness and address problems (Especially by adopting the "per quarter" model) |
| | | People's communication channels for feedback about MTDP: ministerial correspondence and direct contact |
| | | The most common issues based on people's feedback from MTDP: benefit level (Money value) for preventive checkup, ability to receive more comprehensive service at the same price based on the interview, benefits payable only for one preventative checkup |

[a] CDBS: Children Dental Benefit Schemes

[b] DVA: Department of Veterans'Affairs

[c] MTDP: Medicare Teen Dental Plan

[d] RPD: Representative Public Dentists

[e] DBA: Dental Benefits Act

[f] HIA: Health Insurance Act

[g] DBS: Dental Benefits Scheme

[h] ADA: Australian Dental Association

## Operational rules

Table 1 illustrates three main themes and ten related sub-themes in the category of operational rules. These main themes include consequences of the rules governing how dental services are used and provided, how they respond to the collective rules as the norm for the use and delivery of dental services, and how they respond to the regulations on service use and delivery.

According to the results, the content of the Dental Benefits Act 2008 can either respond to collective rules and norms for service use and delivery and at the same time make a response to regulation on service use and delivery (Table 1). In other words, the content of the Act summarises the advertising norms and other promotion strategies as a marketing mix for improving the utilisation of oral health services and promoting oral health behaviours among particular and vulnerable groups. This can be a defined mechanism of responding to collective rules and norms by the community to deliver/provide and utilise oral health services. At the same time, the Act summarises the concept of requirements for utilisation of dental services by eligible groups, rate of utilisation and provision of dental services based on the population strata, financial mechanism, and geographical access, as well as legal facilitators and barriers for service delivery, provision, and utilisation. This category of the rules also indicates the consequences of the rules governing how the service is used and provided. Two consequences of the rules can be considered from the content of the Act as sub-themes including general rates of use and delivery of services, as well as utilisation and service delivery rates based on specific strata/ geographical areas and public and private sectors.

## Collective rules

Similarly, Table 2 demonstrates the main theme and sub-themes related to the category of collective rules. The main theme that emerged here was "forming the level of the constitution to the enforcement of collective rules", which includes two sub-themes: 'Transforming and adapting the rules from the constitutional level' and 'making the mechanism for monitoring and enforcement of collective rules. These two sub-themes make and form the level of the constitution to the enforcement of collective rules.

**Constitutional rules.** And finally, Table 3 presents a related theme and two sub-themes in the category of constitutional rules. "Legitimising the role of collective level of governance" emerged as the main theme and includes the two sub-themes of "power reference and steward authority" and "rules and plans: creation, modification, enforcement, and external monitoring". Power reference and steward authority by the Ministry of Health at the national level and authorisation of the State and Territory governments along with creation, modification, enforcement, and external monitoring of the rules and plans can legitimise the role at the collective level of governance to the constitutional rules at the politician's macro level (Table 3). This process of creating, modification, enforcing and monitoring of the rules and plans can be implemented and organised by the direct and indirect feedback from the community and service users.

Considering the results of the present document analysis, politicians and policy makers at the macro and government level can affect all three constitutional, collective, and operational rules. People and families and oral health providers could then make change, monitor, and enforce formal and informal rules which can be in each of the three categories of operational, collective, and constitutional rules. These three categories of rules can then affect their actions, decisions, and relations (Fig 2).

As is illustrated in Fig 2, the operational rules at the level of oral health providers, including dental specialists and the population as the end users of oral health services, can be demonstrated as rules in use in a mutual interaction with the collective and the constitutional rules.

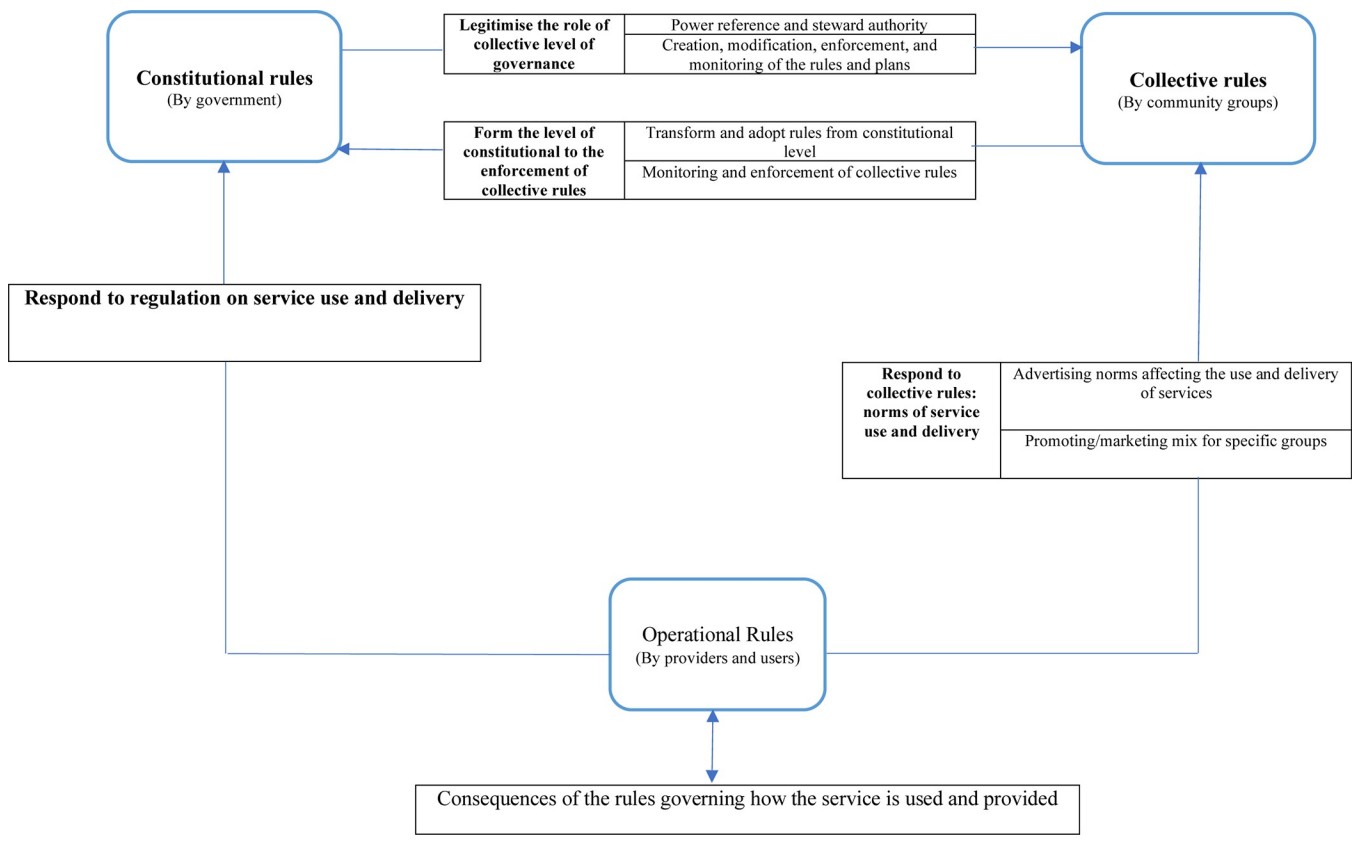

**Fig 2. The framework of rules directed from Dental Benefits Act 2008.**

The consequence of governing the rules at the community level can easily define how the oral health services are really provided and utilised. It is clear that the response is sent to the government level for better regulation of dental service delivery and utilisation. This is then followed with an interaction and advocacy with a diverse range of stakeholders; collaboration across disciplines and capacity to establish, build, and sustain intersectoral and interdisciplinary partnerships with the community groups, non-government sectors, councils, and communities. In this way the rules can be transformed, adopted, monitored, and enforced. Another mechanism of response is occurring from the providers' and users' level and the operational rules to community groups and stakeholders via advertising and promoting the utilisation and provision of oral health services.

## Discussion

This study is conducted to determine how the content of the Dental Benefits Act 2008 can help increase the utilisation of oral health services via two schemes of MTDP and CDBS that the Act mainly focused on during that time. The significance of such an analysis becomes more notifiable and highlighted considering that according to the Australian Institute for Health and Welfare, some groups of the population are at greater risk of poorer oral health, included among them are those with lower socioeconomic status, Australian Aboriginal and Torres Strait Islander people, those who live in regional and remote areas and those who need additional or special health care needs [17].

To achieve the optimal consequences of dental services provision and utilisation, the Act pays attention to a dual mechanism of advertising norms and promoting strategies as a marketing mix for dental services utilisation and at the same time defining the target group via eligibility requirement setting, rates of dental services utilisation and provision based on geographical area / specific strata, financial mechanisms, and the types of bills. Although it seems that the dual mechanism can be considered as the response to collective rules and regulatory and constitutional rules towards the community and stakeholders and the government, it still needs to concentrate on the other significant factors that could help increase the utilisation of dental services among vulnerable groups. The evidence from a retrospective utilisation rate of CDBS as one of the emphasised schemes by the Act also shows that the rate of utilisation during 2014 and 2015 was low and varied among Australian states and territories (18). In contrast with the low rate of utilisation it is important to mention that according to Putri (2020), preventive services were recognised as the most utilised services among the eligible population during 2014–2015 [18]. Such evidence can highlight the gap that although the Act includes the concept of advertisement and promotion as the marketing mix for the schemes, attention was not paid to the concept of social marketing as well in order to promote oral health behaviours. At the same time, in contrast with the inclusion of eligibility requirements for the target groups in the Act, according to the findings of Orr et al. (2021) some lower levels of preventive services utilisation are obvious among Aboriginal children as one of the most eligible and vulnerable groups [19].

The Act has considered two concepts of power reference and steward authority and creation, modification, enforcement, and external monitoring of the rules and plans for legitimising the role of collective level of governance. In this category the concepts of people's communication channels and people's feedback about the schemes is considered. The quantity and quality of the feedback could be diverse according to the results of Nguyen et al. (2020). Those mothers with lower socioeconomic status level, those who have experienced any kind of depression or with lower levels of mental health, and the mothers with a type of health behavioural problem like smoking or even poor oral health behaviours, had the lower tendency to claim the available benefits of CDBS for their children [6]. According to this, it would be important to find the mechanism for weighing the feedback or finding some more robust channels and real feedback on the utilisation of the services by the Australian families.

Another overlooked point that needs more attention in the potential revisions of the Act is to focus more on the coverage of the services highlighted in the Act. For instance, although the present results show that monitoring, transforming, adopting and enforcement of the rules is important to translate them into practice by the end users, evidence implies that some types of preventive services provisions by an alternative workforce are not supported by the fourth revision of the Act such as fluoride varnish items provided by dental assistants because there was not any defined mechanism for receiving and reviewing the changes and revisions via the Act [19]. Similarly, for those Indigenous population groups who use CDBS voucher, two concerns are mentioned. Firstly, the variety of service provision based on different locations that led to a poor coverage of just over 20% of the concession card holders and secondly the exclusion of many services like orthodontics from CDBS and the need for such services to be undertaken by the private sector [20]. So, as the main concentration of the Act and the CDBS is on preventive schemes, it would be more effective and valuable, in the next revisions of the Dental Benefit Act (2008), to include, according to the needs (and unmet needs) of the eligible population, in a package all the probable essential dental services for this population group. Also, from the coverage perspective, there should be a new policy dialogue to identify strategies for increasing the utilisation of the services particularly during and after COVID-19 era. According to Hopcraft and Farmer (2020), from a retrospective analysis of Medicare data on CDBS

utilisation, a large decline was observed in the provision of preventive and diagnostic services during the pandemic in comparison with the utilisation of endodontic and oral surgery services [21].

This study attempted to integrate the perspective of politicians with those of policy makers to reconsider the role and significance of the rules based on the triple collaborations among oral health users and oral service providers, the community, and the stakeholders as well as the government. The Australian Dental Association is among one of main stakeholders which could influence in shaping the rules and service codes. This national voluntary professional organisation which has branches in all the States and Territories, could act as a bridge between the dentistry society particularly private practitioners and the community to help improve the provision of dental services according to the oral health and general health needs of the population.

This study was accompanied with some limitations as follows: the national document analysis of the Acts and legislations would have been enhanced if it were triangulated with an in-depth qualitative data collection (and analysis) from the perspective and experiences of service users and providers, as well as the inclusion of quantitative statistics for the utilisation rate of the schemes. Undertaking a mix-method study for a holistic view prior to the further revision of the Act or probable upcoming schemes are recommended.

## Conclusion

Utilisation of the oral health services by the population is a strategy considered by the Australian health system as a way of achieving, as a final goal, optimum oral health for its population. For this purpose, coordination and integration of the laws and legislations at the Parliament and politician level with those applied policies, interventions, schemes and regulations by the government and policy makers is necessary. The Dental Benefit Act 2008, as a foundation for MTDB and CDBS, is the key Australian legislation which tries to consider all the operational, collective, and constitutional rules with a mutual relationship with the population as the end user of oral health services; the oral health providers; the community and stakeholders; and the government and oral health policymakers. However, a comprehensive attention is still needed in future revisions of the Act according to the contextual factors, socioeconomic and geographical attributes of the population for better implementation of de facto rules and more effective outcomes of the interventions. It is recommended that further research be undertaken utilising a mix-method approach for a holistic view prior to further revisions of the Act or proposal of probable upcoming schemes.

## Acknowledgments

The authors would like to acknowledge and thank A/Prof Mohammad Amin Bahrami who was the external qualitative research expert that reviewed the whole research process, and participated in a debriefing session.

## Author Contributions

**Conceptualization:** Peivand Bastani, Ajesh George, Loc Do.

**Data curation:** Reyhane Izadi.

**Formal analysis:** Reyhane Izadi, Diep Ha.

**Methodology:** Peivand Bastani.

**Project administration:** Nithin Manchery.

**Software:** Reyhane Izadi.

**Supervision:** Loc Do.

**Validation:** Hanny Calache.

**Writing – original draft:** Nithin Manchery, Diep Ha.

**Writing – review & editing:** Peivand Bastani, Hanny Calache, Ajesh George.

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
