## [Decision Letter · Decision Letter 0]

4 Oct 2022

PONE-D-22-22909HOW DOES THE DENTAL BENEFITS ACT ENCOURAGE AUSTRALIAN FAMILIES TO SEEK AND UTILISE ORAL HEALTH SERVICES?PLOS ONE

Dear Dr. Bastani,

Thank you for submitting your manuscript to PLOS ONE. After careful consideration, we feel that it has merit but does not fully meet PLOS ONE’s publication criteria as it currently stands. Therefore, we invite you to submit a revised version of the manuscript that addresses the points raised during the review process.

We look forward to receiving your revised manuscript.

Kind regards,

Rong Zhu, Ph.D.

Academic Editor

PLOS ONE

Journal Requirements:

Reviewers' comments:

Reviewer's Responses to Questions

**Comments to the Author**

1. Is the manuscript technically sound, and do the data support the conclusions?

Reviewer #1: Yes

2. Has the statistical analysis been performed appropriately and rigorously? 

Reviewer #1: N/A

3. Have the authors made all data underlying the findings in their manuscript fully available?

Reviewer #1: Yes

4. Is the manuscript presented in an intelligible fashion and written in standard English?

Reviewer #1: No

5. Review Comments to the Author

Reviewer #1: Dear Authors,

I found this paper interesting to read. A major influence that was not explored at all is the influence/implication of the Australian Dental Association in shaping the rules and the service codes. This is a critical area of concern, and has been an exception to oral health. In many other medical health fields, they are developed by the Department of Health's delegate authority, the Medical Services Advisory Committee, in consultation with stakeholders. Whilst contentious, the paper should explore in more detail the role of the Australian Dental Association with the Dental Benefits Act 2008, and how this influence may be pervasive in achieving the public health goals to ensure universal access to affordable oral health care.

I have provided comments throughout the paper. At times, content was hard too read, and requires extensive grammar editing and sentencing corrections.

Kind regards,

Peer-reviewer

6. PLOS authors have the option to publish the peer review history of their article (what does this mean?). If published, this will include your full peer review and any attached files.

Reviewer #1: No

---

## [Author Response · Author response to Decision Letter 0]

17 Oct 2022

Dear editor

Thank you so much for considering the article for peer review at PLOS One journal. The followings are line by line responses to the respected reviewers’ comments as well as the mentioned corrections in track change in the manuscript’s body:

Background

Page 9

1- The Commonwealth does not provide services; they fund the State and Territories to provide them.

We have omitted any mention of “Commonwealth” in the manuscript as per the reviewer’s comment. 

2- Cost-effective is incorrect, it is effectiveness. No studies have evaluated if they were cost-effective because there are no existing outcomes to make this comparison.

We have corrected “cost-effective” to “effectiveness” in the manuscript according to the reviewer’s comment. 

3- “Due to indexation”

This additional point has been mentioned and clarified the manuscript.

4- There are two different issues here that needs to be separated. Utilisation refers to access, whereas the prevalence of moderate stages of dental caries based on the recent child oral health survey is irrelevant to the CDBS because this program only covers a fraction of the child population. Please address

Thank you for pointing this out. We have revised this section to focus only on the utilisation of the CDBS program to improve clarity.

5- Please reference more recent figures for utilisation of the CDBS, from the Fourth Review of the Dental Benefits Act 2008 or is this figure to the MTDP? Again, this paragraph needs clarity. I note the reference used is the child oral health survey, which does not tell us if children use the CDBS or MTDP.

As suggested, we have included a more recent reference for the utilisation of the CDBS and the concerns of MTDP program.

6- 'Given the challenges for increasing the utilisation of the CDBS... etc...' This forms the basis of this paper.

Thank you for this suggestion. We have incorporated this feedback and improved clarity of this section.

7- Add reference 

A Reference has been added. 

Page 10

8- Sentences appear incomplete? I suspect outcome on utilisation, please clarify.

We have clarified the sentence. 

Methods 

Page 10

9- Reference year required 

We have included all reference years.

10- 2008

We have corrected this to “ACT 2008”

11- Who did this? Please confirm if an author or someone else and include in Acknowledgments.

We have clarified that an expert outside the research team acted as the qualitative research expert and have included further details in the acknowledgement section 

12- Some edition and strikethrough texts are mentioned in the body

These edits have been undertaken 

Results 

13- Capital the first letter of table and figures 

These edits have been undertaken 

14- Confirm if this refers to State and Territory governments (Page 6)

Yes, we have confirmed this 

Discussion 

15- Torres Strait Islander People

The word “People” has been added

16- Page 8, Comma, required

This has been addressed

17- Page 8, Break up concepts, sentence too long.

Thank you for this feedback. We have paraphrased the paragraph.

18- Page 8, Again, what does this mean in terms of outcome, cost-effective needs a comparator for improving an outcome unless you are simply trying to save costs.

We have replaced the term ‘cost-effective’ with ‘effective’ 

19- As per above, consider alignment these concepts with the WHO Global Oral Health Action Plan, using the term essential dental services package,

We have used the term “essential dental services package” as suggested by the reviewer

20- I found this paper interesting to read. A major influence that was not explored at all is the influence/implication of the Australian Dental Association in shaping the rules and the service codes. This is a critical area of concern and has been an exception to oral health. In many other medical health fields, they are developed by the Department of Health's delegate authority, the Medical Services Advisory Committee, in consultation with stakeholders. Whilst contentious, the paper should explore in more detail the role of the Australian Dental Association with the Dental Benefits Act 2008, and how this influence may be pervasive in achieving the public health goals to ensure universal access to affordable oral health care.

Thank you for highlighting this important point. We have included additional points around the potential role of the Australian Dental Association in this area as one of the key stakeholders.

21- At times, content was hard to read, and requires extensive grammar editing and sentencing corrections.

We have reviewed the whole manuscript to address grammatical errors and improve clarity

Abbreviations

22- Organization

The abbreviation for WHO has been corrected.

---

## [Editor Report · Decision Letter 1]

21 Oct 2022

HOW DOES THE DENTAL BENEFITS ACT ENCOURAGE AUSTRALIAN FAMILIES TO SEEK AND UTILISE ORAL HEALTH SERVICES?

PONE-D-22-22909R1

Dear Dr. Bastani,

We’re pleased to inform you that your manuscript has been judged scientifically suitable for publication and will be formally accepted for publication once it meets all outstanding technical requirements.

Kind regards,

Rong Zhu, Ph.D.

Academic Editor

PLOS ONE

---

## [Editor Report · Acceptance letter]

15 Nov 2022

PONE-D-22-22909R1 

How does the dental benefits act encourage Australian families to seek and utilise oral health services? 

Dear Dr. Bastani:

I'm pleased to inform you that your manuscript has been deemed suitable for publication in PLOS ONE. Congratulations! Your manuscript is now with our production department. 

Kind regards, 

on behalf of

Dr. Rong Zhu 

Academic Editor

PLOS ONE